Health status and morphometrics of Galápagos magnificent frigatebirds (Fregata magnificens magnificens) determined by hematology, biochemistry, blood gas, and physical examination

Guzmán Kimberly E. 1
Deresienski Diane 2 3 4 twovets541@gmail.com
http://orcid.org/0000-0002-9792-5653 Muñoz-Pérez Juan Pablo 4 5 6
Passingham Ronald K. 3
Skehel Alice 7
Ulloa Catalina 5
Regalado Cristina 8
http://orcid.org/0000-0003-0716-1387 Lewbart Gregory A. 3 4
http://orcid.org/0000-0002-5523-634X Valle Carlos A. 4 5
1 Oradell Animal Hospital , Paramus, New Jersey , United States
2 College of Veterinary Medicine, Universidad San Francisco de Quito , Quito , Ecuador
3 Clinical Sciences, North Carolina State University , Raleigh, NC , United States
4 Galapagos Science Center, Universidad San Francisco de Quito , Puerto Baquerizo Moreno, Galapagos , Ecuador
5 Colegio de Ciencias Biológicas y Ambientales COCIBA, Universidad San Francisco de Quito USFQ , Quito , Ecuador
6 Biology, University of the Sunshine Coast , Sippy Downs , Australia
7 Biological Sciences, University of the Sunshine Coast , Sippy Downs , Australia
8 Universidad San Francisco de Quito , Quito , Ecuador
Kass Philip
Electronic publication date: 2024 Dec 6
Publication date: 2024
Volume: 12
Electronic Location ID: e18609
Received 2024 Aug 19; Accepted 2024 Nov 7
Copyright: © 2024 Guzmán et al.
Copyright year: 2024
Copyright holder: Guzmán et al.
License: This is an open access article distributed under the terms of the Creative Commons Attribution License, which permits unrestricted use, distribution, reproduction and adaptation in any medium and for any purpose provided that it is properly attributed. For attribution, the original author(s), title, publication source (PeerJ) and either DOI or URL of the article must be cited.
License URL: https://creativecommons.org/licenses/by/4.0/

Keywords: Biochemistry, Fregata magnificens, Magnificent frigatebird, Galápagos Islands, Hematology, Health status, Physical examination, Blood gas

Funding: Heska Corporation Galápagos Academic Institute for the Arts and Sciences GAIAS-Universidad San Francisco de Quito USFQ Galápagos Science Center-USFQ/University of North Carolina-Chapel Hill This research was conducted with support of the Heska Corporation, the Galápagos Academic Institute for the Arts and Sciences GAIAS-Universidad San Francisco de Quito USFQ and the Galápagos Science Center-USFQ/University of North Carolina-Chapel Hill. The funders had no role in study design, data collection and analysis, decision to publish, or preparation of the manuscript.

==============================
The magnificent frigatebird (Fregata magnificens; MFB) is a widely distributed seabird. It has breeding areas in the tropical Atlantic Ocean and the Pacific Ocean (extending along Central America up to Baja California) (Schreiber & Burger, 2001). The Fregata magnificens magnificens (MFB-Gal) subspecies is native to the Galápagos Islands. This is the first-time hematology and blood chemistry parameters have been published for the F. m. magnificens (MFB-Gal) from the Galápagos Islands. Analysis was run on blood samples drawn from n = 16 adult MFB-Gal captured by hand at their nests at North Seymour and Daphne Major Islands in the Galápagos Islands (n = 10 MFB-Gal in June 2017 and n = 6 MFB-Gal in July 2022). There were ten female birds and six male birds in total. A portable blood analyzer (iSTAT) was used to obtain near immediate field results for total carbon dioxide (TCO2), hematocrit (Hct), hemoglobin (Hb), sodium (Na), potassium (K), chloride (Cl), ionized calcium (iCa), total protein (TP), anion gap and glucose. Blood lactate was measured using a portable Lactate Plus™ analyzer. Average heart rate, respiratory rate, body weight, body temperature, biochemistry and hematology parameters were comparable to healthy individuals of other Fregatidae of the same species (magnificent frigatebird subspecies from Brazil, Fregata magnificens, likely F. m. rothschildi) or similar species (great frigatebird, Fregata minor, from the Galápagos Islands). There were some statistically significant differences between the males and females F. m. magnificens (MFB-Gal) in the Galápagos, including bill depth, bill width, wing length, weight, and chloride blood value. The reported results provide baseline data that can be used for comparisons among populations and in detecting changes in health status among Galápagos magnificent frigatebirds and other populations of magnificent frigatebirds.

Introduction

The magnificent frigatebird (Fregata magnificens; MFB) is a tropical bird member of the order Suliformes in the family Fregatidae, previously included in the order Pelecaniformes (Redrobe, 2015). The MFB has two subspecies: the Fregata magnificens magnificens (MFB-Gal; found along the Pacific Ocean) and the Fregata magnificens rothschildi (found along the Pacific and Atlantic Ocean) (Gill, Donsker & Rasmussen, 2022). The MFB is listed as Least Concern under the Red List of Threatened Animals, but its population is decreasing (BirdLife International, 2020). The MFB extend along the Atlantic Ocean from Florida to the Caribbean, the Cape Verde Islands, and the coast of Brazil. Along the Pacific Ocean, the MFB-Gal extend along the coast of the Southern United States, to Mexico, to the Galápagos Islands, Ecuador (Nelson, 1975; Schreiber & Burger, 2001). MFB have a large wingspan, short tarsus, and inadequately waterproof feathers (Schreiber & Burger, 2001; Weimerskirch et al., 2003). MFB fish by themselves, scavenge, kleptoparasitize, and spend most of their time flying (Diamond, 1973; Schreiber & Burger, 2001). The reason for sexual dimorphism (SSD) is debated for MFB, but a theory exists that it could be due to females needing more body reserve to reproduce while males need less body weight to catch prey and male-male competition (Schreiber & Burger, 2001; Tamás, Thomas & Cuthill, 2006).

There are published reports on hematology, blood chemistry, and morphometric values for the great frigatebird (Fregata minor; GFB) species from Isla Genovesa in the Galapagos archipelago (Padilla et al., 2006; Valle et al., 2018). There are published reports on hematology and blood chemistry of the MFB from Rio De Janeiro, Brazil (Scarelli et al., 2020) and MFB from the Southeastern Coast of Brazil (Ferioli et al., 2024). Morphometrics, such as body and length and weight, of MFB around Brazil were reported in Scarelli et al. (2020). No morphometric data was reported for MFB around Brazil in Ferioli et al. (2024). Blood work of rehabilitated MFB from Brazil (not stating which subspecies, but likely the F.m. rothschildi) was reported in Scarelli et al. (2020) and Ferioli et al. (2024), which can be helpful to assess the MFB around Brazils’ health status. This study is the first hematology and blood chemistry for the MFB-Gal (Fregata magnificens magnificens) from the Galápagos Islands.

Seabird health in the Galápagos Islands can be affected by weather, food source availability, natural and man-made disasters, introduced species, and pathogens (Wikelski et al., 2004; Bastien et al., 2014; Rivera-Parra, Levin & Parker, 2014; Rivera-Parra et al., 2015). It is important to document the health parameters of wild animal populations to establish and monitor baseline health status, especially in a population decline (Alonso-Alvarez, 2005; Lewbart et al., 2014, 2018, 2019; Valle et al., 2018, 2020a, 2020b; Tucker-Retter et al., 2021). The Galápagos Islands are an isolated archipelago. We chose to study the MFB-Gal to be able to better understand seabird differences (including genetic differences), especially since the MFB-Gal have isolated breeding and living sites in the Galápagos. The impact of climate change is also an important reason to perform health assessments to understand how animals like seabirds are impacted (Altizer et al., 2013; Dueñas, Jiménez-Uzcátegui & Bosker, 2021; Jiménez-Uzcátegui et al., 2019; Lynton-Jenkins et al., 2021; Muñoz-Pérez et al., 2023). The objective of this study was to describe baseline vital, hematological, blood biochemical, and morphometric parameters for the MFB-Gal (F. m. magnificens) and compare them to previously published data on MFB around Brazil (Scarelli et al., 2020) and GFB from the Galápagos Islands, specifically from North Seymour island and Punta Pitt at San Cristobal island (Valle et al., 2018). This data will be a helpful comparison if there is ever a threat to the population by pathogens, climate change effects, pollution effects, or other anthropogenic factors (Padilla et al., 2006; Padilla & Parker, 2008; Levin & Parker, 2012; Bastien et al., 2014).

Materials and Methods

Ethics statement

The study was performed as part of a population health assessment authorized by the Galapagos National Park Service (permit No. PC-59-17 to C.A. Valle; No. PC-04-22 to G.A. Lewbart) and approved by the Universidad San Francisco de Quito ethics and animal handling protocol. All handling and sampling procedures were consistent with standard vertebrate protocols and veterinary practices.

Study area

Data were collected at Daphne Major Island (0° 25′ 20″ S, 90° 22′ 17″ W) and North Seymour Island (0° 23′ 34″ S, 90° 16′ 56″ W), two of the thirteen MFB-Gal breeding colonies in the Galápagos archipelago. The North Seymour colony has about 130 breeding pairs and the Daphne Major colony has about 93 breeding pairs as of June 2017 (C.A. Valle, 2022, personal communication).

Sampling

In June 2017, four adult MFB-Gal from North Seymour and six adult MFB-Gal from Daphne Major were captured from the nest one at a time, sampled, and examined before the next bird was selected. In July 2022, five adult MFB-Gal from North Seymour and one adult MFB-Gal from Daphne Major were captured one at a time from the nest, sampled, and examined before the next bird was selected. A group of five to six people were involved in each field day. Each morning, each bird was approached slowly and carefully to their nest. The person who captured each bird ducted down slowly and a cane was placed above each birds’ head. Each bird was then captured by their beak. Another person then came in to grab their body, making sure to bring each birds’ wings into their body to decrease risk of fracture. Blood sampling was performed first, then the animals were weighed, standard avian measurements were taken, and heart rate and respiratory rate recorded (Tables 1–3). To minimize the potential effects of handling on blood parameters, sample times were held to about 10 min or less. All the birds were deemed clinically healthy based on their behavior, response to handling, and physical examination by veterinarians. To ensure that individuals were not captured multiple times, a small black sharpie mark was placed on the maxilla of their upper beak to identify that they had already been sampled. Each bird was placed back to the nest or area they were captured from. This method has been used in previous population health assessments (Valle et al., 2018).

Table 1 Morphological and physiological measurements.

Summary of morphological and physiological measurements with time to obtain blood samples for female and male magnificent frigatebirds (Fregata magnificens magnificens; MFB-Gal) at the Galápagos Islands, Ecuador (at Daphne Major and North Seymour), magnificent frigatebirds (Fregata magnificens) in Brazil (Rio de Janeiro), and great frigatebirds (Fregata minor) at the Galápagos Islands, Ecuador (at North Seymour). Galápagos MFB-Gal: Respiratory rate n = 9 because one female was performing a gular flutter due to elevated environmental temperature. [mean ± standard deviation (min-max)].

Parameter	Galápagos MFB-gal female (n = 10)	Galápagos MFB-gal male (n = 6)	Brazil MFBa female (n = 36)	Brazil MFBa male (n = 22)	Galápagos GFBb female (n = 12)	Galápagos GFBb male (n = 18)	
Bill length (mm)	118.08 ± 13.16 (102.99–134.03)	106.78 ± 3.97 (100.85–112.73)	—	—	109.15 ± 3.96 (101–115)	98.28 ± 4.02 (92–106)	
Bill depth (mm)c	26.86 ± 1.01 (25.48–28.38)	24.53 ± 1.33 (22.48–25.93)	—	—	24.85 ± 1.38 (22–26)	23.50 ± 1.89 (20–27)	
Bill width (mm)c	29.75 ± 0.5 (28.88–30.66)	26.75 ± 1.14 (25.37–28.23)	—	—	28.66 ± 1.84 (26–33)	26.7 ± 1.78 (22–29)	
Wing length (mm)c	691.94 ± 12.27 (670.7–710)	642.24 ± 19.18 (610.5–660.6) n = 5	—	—	611.4 ± 14.9 (590–640)	581.6 ± 46.4 (420–620)	
Tarsus length (mm)	26.32 ± 3.25 (21.31–30.94)	27.1 ± 3.28 (22.23–30.99)	—	—	22.8 ± 2.8 (17–26)	23.5 ± 4.9 (1–4)	
Weight (kg)c	1.52 ± 0.16 (1.26–1.78)	1.12 ± 0.11 (0.98–1.25)	1.45 ± 0.16 (1.00–1.80)	1.18 ± 0.16 (1.00–1.70)	1.49 ± 0.12 (1.2–1.6)	1.19 ± 0.09 (1–1.3)	
Total body length (cm)	—	—	87.0–105.0	85.0–101.3	—	—	
Heart rate (beats/min)	301.2 ± 51.85 (200–400)	328.33 ± 51.45 (276–400)	—	—	194 ± 21.94 (152–224)	197.11 ± 28.14 (148–248)	
Respiratory rate (breaths/min)	38.44 ± 13.18 (20–64) n = 9	36.67 ± 12.18 (20–50)	—	—	26 ± 5.25 (20–32)	24.44 ± 6.27 (13–36)	
Sample time (min)	4.3 ± 1.64 (1–6)	3.33 ± 1.03 (2–5)	—	—	4.83 ± 1.74 (3–8)	5.52 ± 2.78 (3–14)	
Handling time (min)	14.5 ± 4.87 (10–24) n = 8	10.6 ± 2.79 (8–14) n = 5	—	—	12.58 ± 1.97 (10–17)	13.77 ± 5.54 (8–29)	
Notes:

a Originally reported by Scarelli et al. (2020).

b Originally reported by Valle et al. (2018).

c Indicates statistically significant differences (P < 0.05) between sexes in F. m. magnificens.

Table 2 Galápagos and Brazil MFB blood comparisons.

Blood gas, biochemical, and hematology values for magnificent frigatebirds (Fregata magnificens magnificens; MFB-Gal) at the Galápagos Islands, Ecuador (at Daphne Major and North Seymour) and magnificent frigatebirds (Fregata magnificens) in Brazil (Rio de Janeiro). Galápagos MFB-Gal: iSTAT Portable Clinical Analyzer and portable Lactate PlusTM analyzer were used; total protein average: two readings were averaged. (mean ± standard deviation (min-max)).

Parameter	Galápagos MFB-Gal (males and females, n = 16)	Brazil MFBa (males and females; n = 58)	
HCTiSTAT (%)	38.89 ± 3.28 (35–46)	48.53 ± 8.76 (31–65)b	
Total proteiniSTAT (g/dL)	5.24 ± 0.9 (3.9–7.6)	4.33 ± 0.14 (4.0–4.6)c	
Hemoglobin (g/dL)	13.21 ± 1.12 (11.9–15.6)	14.08 ± 0.65 (12–15)	
TCO2 (mmol/L)	20.87 ± 1.97 (18–27)	—	
Cl (mmol/L)d	115.44 ± 1.89 (111-119)	—	
Na (mmol/L)	144.4 ± 2.93 (135–148)	—	
K (mmol/L)	4.11 ± 0.79 (2.2–5.2)	—	
iCa (mmol/L)	1.07 ± 0.14 (0.7–1.28)	—	
Glucose (mmol/L)	240.1 ± 18.41 (202–262)	254.33 ± 6.62 (241–267)	
Lactate (mmol/L)	5.06 ± 1.66 (3.1–8.8)	—	
Anion gap	13.37 ± 3.32 (4–18)	—	
Cell types			
Heterophil %	45.09 ± 5.57 (37.5–58)	—	
Lymphocyte%	39.21 ± 8.98 (27–56)	—	
Monocyte %	1.32 ± 1.21 (0–4)	—	
Eosinophil %	14.73 ± 9.77 (1–29)	—	
Basophil %	0.11 ± 0.29 (0–1)	—	
Estimated WBC (103/µL)	12.63 ± 2.69 (6.8–15.75)	14.79 ± 0.99 (12.85–16.73)	
Calculated absolute values of cell types			
Heterophil (ABS) (103/µL)	5.65 ± 1.27 (3.74–8)	7.93 ± 0.76 (6.45–9.41)	
Lymphocyte (ABS) (103/µL)	4.84 ± 1.04 (2.72–6.64)	3.83 ± 0.5 (2.85–4.80)	
Monocyte (ABS) (103/µL)	0.16 ± 0.14 (0–0.46)	1.60 ± 0.25 (1.11–2.1)	
Eosinophil (ABS) (103/µL)	2.02 ± 1.41 (0.09–4.25)	0.31 ± 0.06 (0.18–0.43)	
Basophil (ABS) (103/µL)	0.0095 ± 0.027 (0–0.085)	0.39 ± 0.14 (0.11–0.66)	
Notes:

a Originally reported by Scarelli et al. (2020).

b Packed Cell Volume (PCV) used.

c Portable refractometer used.

d Indicates statistically significant differences (P < 0.05) between sexes in F. m. magnificens.

Table 3 MFB and GFB blood value comparisons.

Blood gas, biochemical, and hematology values for magnificent frigatebirds (Fregata magnificens magnificens; MFB-Gal) at the Galápagos Islands, Ecuador (at Daphne Major and North Seymour) and great frigatebirds (Fregata minor) at the Galápagos Islands, Ecuador (at North Seymour). Galápagos MFB-Gal: iSTAT Portable Clinical Analyzer and portable Lactate PlusTM analyzer were used; total protein average: two readings were averaged. (mean ± standard deviation (min-max)).

Parameter	Galápagos MFB-Gal (males and females, n = 16)	Galápagos GFBa (males and females, n = 30)	
HCTiSTAT (%)	38.89 ± 3.28 (35–46)	36.68 ± 3.08 (28–42)	
Total proteiniSTAT (g/dL)	5.24 ± 0.9 (3.9–7.6)	4.89 ± 1.06 (1.3–6.2)	
Hemoglobin (g/dL)	13.21 ± 1.12 (11.9–15.6)	12.47 ± 1.05 (9.5–14.3)	
TCO2 (mmol/L)	20.87 ± 1.97 (18–27)	22.50 ± 3.93 (15–33)	
Cl (mmol/L)b	115.44 ± 1.89 (111–119)	112.41 ± 5.25 (100–123)	
Na (mmol/L)	144.4 ± 2.93 (135–148)	142.43 ± 3.30 (134–148)	
K (mmol/L)	4.11 ± 0.79 (2.2–5.2)	3.66 ± 0.45 (2–4)	
iCa (mmol/L)	1.07 ± 0.14 (0.7–1.28)	1.10 ± 0.18 (0.57–1.3)	
Glucose (mmol/L)	240.1 ± 18.41 (202–262)	260.35 ± 34.75 (171–330)	
Lactate (mmol/L)	5.06 ± 1.66 (3.1–8.8)	3.65 ± 1.05 (1–7)	
Anion gap	13.37 ± 3.32 (4–18)	12.44 ± 3.68 (2–18)	
Cell types			
Heterophil %	45.09 ± 5.57 (37.5–58)	41.72 ± 3.60 (34–51)	
Lymphocyte%	39.21 ± 8.98 (27–56)	36.83 ± 4.05 (30–46)	
Monocyte %	1.32 ± 1.21 (0–4)	0.31 ± 0.55 (0–2)	
Eosinophil %	14.73 ± 9.77 (1–29)	21.14 ± 4.43 (8–29)	
Basophil %	0.11 ± 0.29 (0–1)	0 ± 0	
Estimated WBC (103/µL)	12.63 ± 2.69 (6.8–15.75)	—	
Calculated absolute values of cell types			
Heterophil (ABS) (103/µL)	5.65 ± 1.27 (3.74–8)	—	
Lymphocyte (ABS) (103/µL)	4.84 ± 1.04 (2.72–6.64)	—	
Monocyte (ABS) (103/µL)	0.16 ± 0.14 (0–0.46)	—	
Eosinophil (ABS) (103/µL)	2.02 ± 1.41 (0.09–4.25)	—	
Basophil (ABS) (103/µL)	0.0095 ± 0.027 (0–0.085)	—	
Notes:

a Originally reported by Valle et al. (2018).

b Indicates statistically significant differences (P < 0.05) between sexes in F. m. magnificens.

Morphological measurements

Body weight was obtained using a dynamometer (Pesola (R) spring scale Praz¨isions-waagen AG, Schindellegi, Switzerland; 2 kg scale capacity). Bill length, bill depth, bill width, and tarsus length were measured using metric calipers (Truper stainless steel; 6 inch/150 mm digital micrometer waterproof; Zhejiang, China and Mexico). Wing chord was measured using a flexible measuring tape (Pretul Pro-5MEB; 5 m/6 ft; 69 mm, Mexico). The sex was determined according to the species’ adult plumage sexual dimorphism (Diamond, 1973; Schreiber & Burger, 2001).

Blood sample collection and handling

Blood sampling and handling methods were as previously described (Valle et al., 2018).

Blood gas and biochemistry parameters

The blood gas and biochemistry analyses and parameters measured were as previously described (Valle et al., 2018). The veterinary literature is rich with studies that have utilized the iSTAT for avian health assessment from a variety of species (Steinmetz et al., 2007; Harms & Harms, 2012; Rettenmund, Heatley & Russell, 2014; Harter et al., 2015; Ratliff et al., 2014, 2017).

Statistics

Standard descriptive statistics of all parameters were computed for each sex. All parameters were tested for differences in morphometrics, blood chemistry, and hematology between the sexes using the Wilcoxon test (Tables 1–3, and Fig. 1). All statistical analysis was run using R 4.0.4 (R Core Team, 2021), with a standard α level of 0.05.

Figure 1 Significant differences between sexes.

Bill depth, bill width, wing length, weight (in kilograms), and chloride (Cl) box plots for males and females of the Magnificent Frigatebirds (Fregata magnificens magnificens; MFB-Gal) from the Galápagos Islands. Significant differences (P > 0.05) were found between sexes for all parameters displayed here.

Results

Summary statistics of all morphological and physiological measurements as well as the time to obtain blood samples (from capture) are provided in Table 1. Summary statistics of all the blood chemistry and hematology values, including percent and absolute value of cell type, are provided in Tables 2 and 3. Box plots for males and females of the Galápagos magnificent frigatebirds (Fregata magnificens magnificens; MFB-Gal) were made to show the significant parameters between the sexes (Fig. 1). The MFB from Scarelli et al. (2020) and GFB from Valle et al. (2018) obtained from each article were presented in Tables 1–3. The MFB article from Brazil was presented but direct statistical comparison of physiologic measurements cannot be made with healthy birds because the Brazilian birds were admitted to a rehabilitation center for wing injury or signs of exhaustion. Absolute values for each cell type were calculated by multiplying the estimated white blood cell count by each white blood cell type percentage. No blood parasites were found. Manual hematocrit levels could not be accurately recorded for all samples due to remote field conditions. Bill depth, bill width, wing length, weight (kg) and chloride showed significant differences between male and female MFB-Gal (Fig. 1). Mean bill length, wing length, and tarsus length parameters were slightly lower for the Galápagos GFB, when compared to the MFB-Gal (Table 1). Mean heart rate, respiratory rate, and handling time parameters for the Galápagos GFB were lower, when compared to MFB-Gal (Table 1). All other MFB-Gal morphometric parameters were relatively similar when compared to Brazil MFB and/or Galápagos GFB (Table 1). Mean HCT parameters were higher for the Brazil MFB, when compared to the MFB-Gal (Table 2). Mean eosinophils parameters for the Brazil MFB were lower, when compared to MFB-Gal (Table 2). Mean estimated WBC, heterophil, monocyte, and basophil parameters for the Brazil MFB were higher, when compared to MFB-Gal (Table 2). All other MFB-Gal blood chemistry and hematology parameters were relatively similar when compared to Brazil MFB (Table 2). Mean glucose and eosinophil % parameters were higher for the Galápagos GFB, when compared to the MFB-Gal (Table 3). Mean lactate and heterophil % parameters for the Galápagos GFB were lower, when compared to MFB-Gal (Table 3). All other MFB-Gal parameters were relatively similar when compared to Galápagos GFB blood chemistry and hematology (Table 3).

Discussion

This study reports an in-depth array of morphometric, vital, blood biochemical, and hematological parameters from Galápagos magnificent frigatebirds (Fregata magnificens magnificens; MFB-Gal). Health studies on MFB-Gal from the Galápagos Islands has not been done before. This is the first time MFB-Gal data from the Galápagos archipelago has been collected and presented for publication. The MFB-Gal from the Galápagos, MFB from Scarelli et al. (2020) from Brazil, and the GFB described in Valle et al. (2018) from the Galápagos Islands were presented. The MFB from Scarelli was included to help inform clinical decisions of all the available health data on the MFBs. We were interested to see if the morphometric measurements, physiological measurements, blood chemistry values, and hematology values were comparable (see Tables 1–3).

MFB morphometric values, such as the mean body weight in males and females from Brazil were similar to the mean body weight in males and females from the Galápagos Islands (MFB-Gal) (Table 1). Mean weight of female MFB from Brazil was 1.45 kg while mean body weight of female MFB-Gal from Galápagos was 1.52 kg. Mean body weight of male MFB from Brazil was 1.18 kg while mean body weight of male MFB-Gal from Galápagos was 1.12 kg. Length of each bird from Brazil and the Galápagos Islands was measured differently. The Brazilian study measured the length of each bird. The Galápagos Islands MFB-Gal study used typical ornithological bird biometric measurements such as wing length, tarsal length, bill length, bill depth and bill width, therefore size comparison is difficult to make between the Brazil and Galapagos MFB-Gal. Female Galápagos MFB-Gal have a greater mean body weight, wing length, bill depth, bill width, and bill length than male Galápagos MFB-Gal (statistically significant for body weight, bill depth, bill width, and wing length). Bill depth, bill width, wing length, and weight of male and female MFB-Gal showed significant differences, likely due to their sexual dimorphism (Fig. 1). GFB-Gal bill length, bill depth, and bill width from Galápagos are shorter than the same parameters in MFB-Gal from Galápagos.

Heart rate and respiratory rate in male and female Galápagos MFB-Gal was faster (male mean heart rate 328.33 beats per min, female mean heart rate 301.2 beats per min; male mean respiratory rate 36.67 breaths per min, female mean respiratory rate 38.44 breaths per min) when compared to male and female Galápagos GFB (male mean heart rate 197.11 beats per min, female mean heart rate 194 beats per min; male mean respiratory rate 24.44 breaths per min, female mean respiratory rate 26 breaths per min) (Table 1). The slower heart rate in female MFB-Gal from the Galápagos Islands has been seen in other avian species, particularly due to their larger body weight and size (Calder, 1968, Lindstedt & Calder, 1976, 1981). The mean handling time for MFB-Gal from the Galápagos Islands (14.5 min for females and 10.6 min for male) was comparable to the mean handling time for GFB (12.58 min for females and 13.77 min for male) (Table 1). The blood sampling time for MFB-Gal from the Galápagos Islands (4.3 min for females and 3.33 min for male) was also comparable to the blood sampling time for GFB (4.83 min for females and 5.52 min for male) (Table 1).

Blood gas, biochemistry, and hematology mean values for female and male MFB-Gal from the Galápagos Islands were overall comparable. The significant difference of the blood value chloride between male and female MFB-Gal show females have a slightly lower chloride level (Fig. 1). The hematocrit (HCT %) for the female MFB-Gal from the Galápagos Islands (mean female HCT 38.1%, mean male HCT 39.67%) was lower than the MFB from Brazil (mean female HCT 51.25%, mean male HCT 38.23%), otherwise overall comparable. This could be due to the Galápagos hematocrit samples coming from the iSTAT data and Brazil samples coming from the manual centrifuge method, the iSTAT usually producing a lower hematocrit (Wolf, Harms & Beasley, 2008; Rettenmund, Heatley & Russell, 2014). The GFB from the Galápagos Islands had similar results, showing iSTAT hematocrit values lower than manual hematocrit values (Valle et al., 2018). Hemoglobin values were comparable for the MFB-Gal from the Galápagos Islands and Brazil. The mean total protein values were slightly higher for the MFB-Gal from the Galápagos Islands (mean female 5.15 g/dL, mean male 5.32 g/dL) compared to the MFB from Brazil (mean female 4.37 g/dL, mean male 4.26 g/dL). The mean potassium for male MFB-Gal from the Galápagos Islands (4.22 mmol/L) was higher than the mean potassium from GFB (3.77 mmol/L) from the Galápagos Islands, which can be due to a sampling artifact from release of potassium during clotting. The mean glucose for MFB-Gal from the Galápagos Islands (mean female 244.7 mmol/L, mean male 235.5 mmol/L) was slightly lower than the mean glucose of MFB from Brazil (mean female 259.69 mg/dL, mean male 245.55 mg/dL), likely due to the Brazil MFB having an increased level of stress since they were in need of physical rehabilitation. The mean lactate levels of the MFB-Gal from the Galápagos Islands (mean female 4.34 mmol/L, mean male 5.78 mmol/L) were higher than the GFB-Gal from the Galápagos Islands (mean female 3.98 mmol/L, mean male 3.31 mmol/L). The mean TCO2, Cl, Na, most K values, anion gap, and iCa MFB-Gal values from the Galápagos Islands were comparable to the GFB from the Galápagos Islands.

The estimated mean white blood cell values for the MFB-Gal from the Galápagos Islands (mean female 12.78 103/µL, mean male 12.47 103/µL) were slightly lower compared to the values for MFB from Brazil (mean female 15.76 103/µL, mean male 13.20 103/µL) (Table 2). The mean of the absolute heterophils and monocytes of the MFB-Gal from the Galápagos Islands (absolute heterophil mean female 5.64 103/µL, mean male 5.66 103/µL; absolute monocytes mean female 0.09 103/µL, mean male 0.22 103/µL) were lower compared to the values for MFB from Brazil (absolute heterophil mean female 7.18 103/µL, mean male 9.17 103/µL; absolute monocytes mean female 1.59 103/µL, mean male 1.63 103/µL). The slightly higher white blood cell values, absolute heterophils, and absolute monocytes for the Brazilian MFB could be possibly due to wing injury or inflammation. The mean of the absolute lymphocytes of the MFB-Gal from the Galápagos Islands (mean female 4.56 103/µL, mean male 5.11 103/µL) were slightly higher when compared to the values for MFB from Brazil (mean female 3.94 103/µL, mean male 3.64 103/µL). The mean of the absolute eosinophils of the MFB-Gal from the Galápagos Islands (mean female 2.52 103/µL, mean male 1.52 103/µL) were higher compared to the values for MFB from Brazil (mean female 0.398 103/µL, mean male 0.33 103/µL). The mean of the absolute basophils of the MFB-Gal from the Galápagos Islands (mean female 0.005 103/µL, mean male 0.014 103/µL) were slightly lower compared to the values for MFB from Brazil (mean female 0.27 103/µL, mean male 0.58 103/µL), similar to the absolute eosinophils values for the Brazil and Galápagos Islands MFB. Obtaining a higher sample size would be helpful to have a better standard for morphological, physiological measurements, blood gas, biochemical, and hematology values. However, care must be taken when capturing adult MFB from nests with young MFB or eggs due to aggressive kleptoparasitism from other MFB’s in the colony. Since we could not sample MFB-Gal with young or eggs in their nests, our studies’ ability to capture a large number of MFB-Gal was quite limited. Further work can be done to sample MFB-Gal during different times of year and regions to obtain a larger data base and better understanding of their health.

Tourism in the Galápagos continues to increase and may have an effect on the health of Galápagos seabirds (De Groot, 1983; Epler, 2007; Toral-Granda et al., 2017; Walsh & Mena, 2016). It is important to conduct health assessments to monitor the health of seabird populations in the Galápagos Islands. Knowing the healthy normal values of seabird species such as Galapagos magnificent frigatebirds (Fregata magnificens magnificens; MFB-Gal) is paramount to making accurate health assessments.

Supplemental Information

Supplemental Information 1 2017 raw health data.

Details of the various health parameters assayed.

Supplemental Information 2 2022 Health Data.

Details of the various parameters from 2022.

The authors thank David Ascencio, Maryuri Yepez, Diego Quiroga, Carlos Mena, Stephen Walsh, Philip Page, Jason Castaneda, Tillie Laws, and Michael Levy for support and assistance in this project.

Additional Information and Declarations

Competing Interests

Author Contributions

Animal Ethics

Field Study Permissions

Data Availability

The authors declare that they have no competing interests.

Kimberly E. Guzmán performed the experiments, analyzed the data, prepared figures and/or tables, authored or reviewed drafts of the article, and approved the final draft.

Diane Deresienski conceived and designed the experiments, performed the experiments, analyzed the data, prepared figures and/or tables, authored or reviewed drafts of the article, and approved the final draft.

Juan Pablo Muñoz-Pérez performed the experiments, authored or reviewed drafts of the article, and approved the final draft.

Ronald K. Passingham performed the experiments, authored or reviewed drafts of the article, and approved the final draft.

Alice Skehel performed the experiments, authored or reviewed drafts of the article, and approved the final draft.

Catalina Ulloa conceived and designed the experiments, performed the experiments, authored or reviewed drafts of the article, and approved the final draft.

Cristina Regalado performed the experiments, authored or reviewed drafts of the article, and approved the final draft.

Gregory A. Lewbart conceived and designed the experiments, performed the experiments, analyzed the data, authored or reviewed drafts of the article, and approved the final draft.

Carlos A. Valle conceived and designed the experiments, performed the experiments, analyzed the data, prepared figures and/or tables, authored or reviewed drafts of the article, and approved the final draft.

The following information was supplied relating to ethical approvals (i.e., approving body and any reference numbers):

The study was performed as part of a population health assessment authorized by the Galapagos National Park Service (permit No. PC-59-17 to C.A. Valle; No. PC-04-22 to G.A. Lewbart) and approved by the Universidad San Francisco de Quito ethics and animal handling protocol. All handling and sampling procedures were consistent with standard vertebrate protocols and veterinary practices.

The following information was supplied relating to field study approvals (i.e., approving body and any reference numbers):

The Galapagos National Park issued research permits in 2017 and 2022 for this field work.

The following information was supplied regarding data availability:

The raw data is available in the Supplemental Files.

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
