# Peer review of "Health status and morphometrics of Galápagos magnificent frigatebirds (Fregata magnificens magnificens) determined by hematology, biochemistry, blood gas, and physical examination"

_PeerJ, doi:10.7717/peerj.18609_

## Round 0.1 · original submission · Major Revisions

· Academic Editor

Major Revisions

Please address every comment by the reviewers, including those included in the attached annotated reviewed manuscript. In composing your rebuttal, please list each comment, your response, and how the manuscript has been changed.

Reviewer 1 ·

Basic reporting

The overall structure is good, but the objectives, justifications and presentation of the state of knowledge about the health of this species need to be improved, as the reader will only understand some issues in the discussion.

Experimental design

The study is very important and deserves to be published, but the title and introduction need to make the state of knowledge on the topic addressed and the reasons for choosing Galapagos birds for the study clearer. It is also important to mention the subspecies that occur in Galapagos, and this is an extremely important issue, since the Galapagos has a different subspecies than Brazil, where the data were compared. Therefore, the authors need to make this very clear in the introduction, so that the comparison can be better understood.

Validity of the findings

The findings are interesting, but it is necessary to mention the subspecies in question, and consider the comparisons.

Additional comments

It is necessary to reword the manuscript

Annotated reviews are not available for download in order to protect the identity of reviewers who chose to remain anonymous.

·

Basic reporting

The table titles seem excessively long, and much of the information they contain would be more appropriately placed within the results section.

Experimental design

no comment

Validity of the findings

Although the sample size of the study is limited, it is important to enhance the scientific validity of the findings by applying appropriate statistical analyses, such as an ANOVA test. It would be advisable to focus the discussion on the indicators that demonstrate statistically significant differences, while the remaining data can either be presented in the results section or reported in the tables.

Additional comments

This article provides a comprehensive examination of the health status of Magnificent Frigatebirds (Fregata magnificens) from the Galápagos Islands, utilizing haematological and biochemical parameters, blood gas composition analysis, and physical examinations.
Overall, the study represents a valuable addition to the body of knowledge on seabird health and holds practical relevance for conservation efforts, as well as for veterinarians involved in the rehabilitation and management of these birds in breeding centers.

---

## Round 0.2 · accepted · Accept

· Academic Editor

Accept

Thank you for making the requested modifications to your manuscript. We are pleased to accept it for publication.

·

Basic reporting

no comment

Experimental design

no comment

Validity of the findings

no comment

Additional comments

This article represents a solid scientific study that makes a significant contribution to our understanding of the health status of Galápagos Magnificent Frigatebirds. The authors have effectively addressed the comments from the first round of review, resulting in a clear and coherent text. The article is suitable for publication and will appeal to researchers and practitioners in ecology, veterinary medicine, and conservation.